# Exploring the Dzi Bead with Synchrotron Light: XRD, XRF Imaging and μ-XANES Analysis

**Averie Reinhardt [1], Renfei Feng [2]**, **Qunfeng Xiao [2], Yongfeng Hu [2] and Tsun-Kong Sham [1],***

[1] Department of Chemistry, Western University, 1151 Richmond Street, London, ON N6A 5B7, Canada; areinha2@uwo.ca
[2] Canadian Light Source, 44 Innovation Boulevard, Saskatoon, SK S7N 2V3, Canada; Renfei.Feng@lightsource.ca (R.F.); Qunfeng.Xiao@lightsource.ca (Q.X.); Yongfeng.hu@lightsource.ca (Y.H.)
\* Correspondence: tsham@uwo.ca

**Abstract:** The origin of Dzi beads, also called "tian zhu", has always been a mystery. These beads come in a variety of patterns, shapes and sizes. They have cultural and heritage significance in Tibet and areas surrounding the Himalayas. The most recognized beads are those with the "eye" pattern. They are said to ward off evil spirits. Due to their reputation, the demand for Dzi beads has increased in Asia. Herein, we report a study of a Dzi bead with a three-eye pattern using X-ray diffraction (XRD), X-ray fluorescence (XRF), X-ray absorption near edge structure (XANES) and imaging techniques. This is a novel area for Dzi bead research using X-rays from a synchrotron light source to determine the chemical composition of the bead, if the pattern is natural or man-made or if the bead is genuine or a replica. These techniques revealed the bead to be composed of agate (silicon dioxide). An interesting feature on the bead's surface was the etched rings, which were observed to contain regular copper hot spots on their circumference. Our results suggest that the Dzi bead was genuine and started out as an earth-formed agate, with the pattern crafted.

**Keywords:** Dzi bead; agate; X-ray diffraction; X-ray fluorescence; X-ray absorption near edge structure; X-ray imaging

## 1. Introduction

The origin of Tibetan Dzi beads, also known as "tian zhu" (heaven's pearl) or commonly accepted as Tibetan beads, has always been a mystery. They first appeared between 2000 and 1000 BC in countries surrounding the Himalayas. Authentic Dzi beads are found primarily in Tibet, according to Bolin [1]. There are many legends about how these beads came to be. This is because the Tibetan people believe the beads are of supernatural origin [2]. One story tells of semigods owning them as ornaments. It is said that the beads would be thrown away if they became imperfect in any way. This story explains why Dzi beads are never found in perfect condition [1,2]. Another story is perhaps more well-known and widespread, describing the Dzi beads to be insects. The story tells of a man in the mountains who supposedly threw his hat over such an insect, petrifying it. This petrified insect is said to have been a Dzi bead [1,2]. There are a multitude of other stories that have been told, including the Dzi bead being found in the horns of slaughtered animals as well as in dung [1–3]. The people of Tibet hold these beads in high regard and consider them heritage gems; they are reluctant to sell them for low sums, especially to those in the Western world. Therefore, there has been very little scientific research conducted on them. It is generally accepted, however, that Dzi beads are made from agate, a form of chalcedony quartz with a chemical composition of $SiO_2$ [1–3].

It is said that the owner or wearer of a Dzi bead is protected from catastrophe. The Dzi bead supposedly wards off evil spirits that might have ill effects on the wearer [1,2]. While being worn,

if the bead breaks or chips, it is thought to have served its purpose and has absorbed the energy of a catastrophic event that was intended for the wearer. Thus, the effects of the bead are rendered useless, as it is no longer regarded as "pure" [1,2]. The only time a Dzi is deliberately broken is for medicinal purposes. Doctors in Tibet use powder from the beads mixed with herbs to treat ailments such as epilepsy [1,2].

In Ebbinghouse and Winsten's article, they mention three techniques that have been recorded for creating Dzi beads [2]. The first technique for creating a white pattern on a natural stone background involves the painting of an alkali substance onto the surface of the bead and then firing the whole bead [3–5]. The area that has been painted then turns white, and this continues through the surface into the Dzi bead interior [6,7]. The second technique creates a black design on a whitened background. The whitening is done with the aforementioned technique on the entire bead, and then the dark pattern is painted with a chemical such as copper nitrate [8]. Using similar techniques, the third type of Dzi is a black design on a natural stone background [2–8]. It should also be noted that drill bits made of reed in the old days and then copper were used to drill holes in the crafting process. Bolin's book (p. 29) also describes similar treatments in the craft, including darkening with plant sugar and heat, beaching and white line etching with natron (a naturally occurring mixture of sodium carbonate decahydrate, $Na_2CO_3 \cdot 10H_2O$, and sodium bicarbonate, $NaHCO_3$, along with small amounts of sodium chloride and sodium sulfate) and protecting the desired areas with grease, clay, etc. [1].

The Tibetan people only regard certain Dzi beads as real or "pure". Dzi beads come in a varied array of patterns, shapes and sizes. A set of patterns that are universally recognized as "pure" are the "eye" patterns, as shown in Figure 1a,b together with representative fakes (Figure 1c) and the three-eye specimen we used in this investigation, Figure 1d,e. Every pattern holds a different meaning and benefit to the wearer [1–3]. The most highly coveted Dzi bead is the nine-eye bead. This bead symbolizes the nine planets, holds the activity of the entire universe and possesses the wisdom of humanity [1,2], and it is said to assist the owner in gaining wealth, good health, success, power, compassion and glory [1–8].

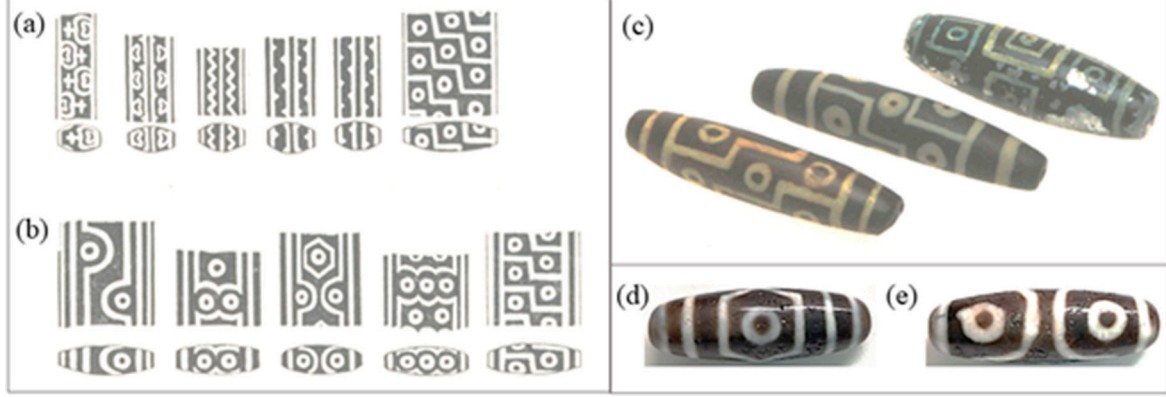

**Figure 1.** Dzi bead patterns. (**a**) The most precious Dzi beads. (**b**) Beads that are of lesser value than (**a**) but are most recognized and worn as traditional ornaments [2]. (**c**) Dzi bead imitations. Left to right: plastic with metal core, polymer clay and painted aluminum [4]. (**d**,**e**) Photograph of the Dzi bead used in this study, showing a one-eye pattern on one side and a two-eye pattern on the opposite side. The middle column in (**b**) depicts what the three-eye bead looks like when unfolded.

Tibetans have created criteria for what makes a Dzi bead valuable. As mentioned, the bead must have a desirable pattern that is considered "pure". A Dzi bead must possess a round cross-sectional shape and not appear thin [1,2]. As noted above, the Dzi bead must not be broken or chipped, as this is said to reflect the fact that it has already served its purpose. When held up to the sun, the bead should be translucent or reveal any internal flaws with the pattern, although there are exceptions [1,2]. Tibetans know how to identify a genuine Dzi bead from a fake; however, imitations have been created because these beads are so valuable.

The demand for Dzi beads in Asian regions has increased and spurred the production of replicas. Some replicas have such minute details that they must be observed more closely under a microscope or cut open to reveal if they are fake. In most cases, it has been identified that a replica weighs less than the original [4]. Other replicas are fitted with a metal center to imitate the weight of the real beads. Dzi bead replicas are usually made of plastic, glass or a light metal such as aluminum; examples are shown in Figure 1c [4].

The objective of this work is to investigate the pattern, the elemental and chemical composition as well as the structure of a three-eye Dzi bead using conventional laboratory techniques, such as optical microscopy and scanning electron microscopy (SEM), and advanced X-ray techniques from a synchrotron source. X-ray diffraction (XRD) reveals the crystal structure and, hence, mineral formation of the bead; X-ray fluorescence (XRF) reveals its elemental composition; and X-ray absorption near edge structure (XANES) reveals the local structure of a selected element of interest. Using a micro-X-ray beam, we can conduct microscopy and microanalysis of the different patterns of the specimen. We have gained considerable experience over the years in the research and development of these techniques [9] and are encouraged by our recent success in retrieving high-resolution images from badly tarnished daguerreotypes [10–12], the first public photographs in human history. The applications of X-rays from synchrotrons in art and archaeology have also been reviewed recently [13]. Based on these techniques, we will attempt to investigate if the specimen is genuine or fake, if the pattern is natural or man-made and if there is any evidence for the crafting methods we noted above, as described by Bolin and Ebbinghouse and Winsten [1,2]. We show below that we can provide some positive answers for these questions in this preliminary study using synchrotron radiation.

## 2. Materials and Methods

### 2.1. The Three-Eye Dzi Bead

The three-eye bead investigated in this work was purchased in a jewelry store in Chengdu, Sichuan, China, and it came with an authenticity certificate (Figure S1). The bead was sliced in the middle, as shown in Figure S2. Slicing was carried out in the Department of Earth Sciences at Western University. A MK Diamond brand tile saw with a diamond embedded blade was used to slice the Dzi bead into several pieces. The bead was cut open to reveal the interior for further imaging. It should be noted that the middle piece was chipped, which is not unexpected from a hard stone like agate. A small piece was ground to power. Both the slices and the powder specimens were used for XRD measurements.

### 2.2. Optical and Scanning Electron Microscope, Energy-Dispersive X-ray Spectroscopy

The optical microscope used was a Mitutoyo WF with a ZEISS Axiocam 1Cc5 color Charge Coupling Device (CCD) camera. The magnifications available were 5×, 10×, 20× and 100×. The Dzi bead was affixed to the glass of the stage with a sticky carbon tab to ensure its stability. The scanning electron microscope was a LEO/ZEISS 1540XB instrument. Both the optical and SEM measurements were made in Western's Nanofabrication Facility. Images were collected from a piece of the Dzi bead, specifically on the outer surface of the bead. Special attention was paid to the etched rings on the surface. The Energy Dispersive X-ray (EDX) detector model was the N-Max 50 by Oxford Instruments. This detector was coupled to the SEM, and the emitted fluorescent X-rays from the sample surface were collected to perform a complete elemental analysis on specific sections on the Dzi bead surface.

### 2.3. XRD

XRD was used to characterize the surface of the light and dark regions; the edge of a slice and a powder sample of the Dzi bead were investigated to determine its structure. Powder XRD was conducted with a Rigaku Ru200BVH diffractometer equipped with a Co $K_\alpha$ ($\lambda = 1.8$ Å) X-ray source. The diffraction patterns were measured over a 2θ range of 2° to 82°. The diffraction patterns of quartz ($SiO_2$), graphite

and copper oxide, CuO, were used as references to compare with the crystal structure of the Dzi bead. The reason for a graphite and copper oxide reference will become clear in the discussion below. XRD of the Dzi bead surface and the edge of a slice was conducted using the VESPERS (Very Sensitive Elemental and Structural Probe Employing Radiation from a Synchrotron) beamline of the Canadian Light Source [14]. It is a hard X-ray beamline equipped with optical elements to deliver a polychromatic beam (pink beam) and multibandpass monochromatic beams (using a double-multilayer and double-crystal monochromator with Si(111) crystals, respectively) as well as microbeam (using a Kirkpatric-Baez (KB) mirror) capabilities for XRD, XRF and imaging [14,15]. Using VESPERS (Very Sensitive Elemental and Structural Probe Employing Radiation from a Synchrotron), XRD was performed with monochromatic X-rays at 8 keV (1.55 Å)~Cu $K_\alpha$ and 12 keV ($\lambda = 1.03$ Å), and the diffracted beam was detected by a Pilatus 1M pixel area detector. The diffraction rings were converted to intensity versus 2θ plots for easy comparison with the references. It should be noted that monochromatic X-rays from the synchrotron are tunable in energy, of a higher energy resolution and at least $10^4$ times brighter than those of laboratory sources, making the technique more desirable for faster acquisition and higher-quality data for Rietveld refinement, and the microbeam (~5 μm) allows for small area diffraction and imaging.

### 2.4. X-Ray Microbeam Analysis: XRF and XANES

XRF and imaging experiments were conducted using the Very Sensitive Elemental and Structural Probe Employing Radiation from a Synchrotron (VESPERS) [14,15] and the Soft X-ray Microanalysis Beamline (Soft X-ray Microanalysis Beamline (SXRMB)) [16,17] beamline of the Canadian Light Source. The former produces monochromatic hard X-rays from 5 to 30 keV, and the latter delivers tender X-rays from 1.7 to 10 keV. With VESPERS, an X-ray energy of 10 keV was selected for XRF excitation, which is above the Cu K-edge (edge: 8979 eV; $K\alpha_1$: 8048 eV), an element of interest in our pursuit, providing a good cross-section and, hence, fluorescence yield from other first-row transition elements at the K-edge, such as Fe (edge: 7112 eV; $K\alpha_1$: 6404 eV). The SXRMB beamline is equipped with a double-crystal monochromator with dual crystal sets of InSb(111) and Si(111) crystals. The former is used for Si K-edge studies with a slightly better photon throughput. It is also equipped with a tender X-ray microprobe endstation that is capable of tracking elements such as Si (K-edge: 1840 eV; $K\alpha_1$ at 1740 eV), K (K-edge: 3608 eV; $K_{\alpha1}$: 3314 eV) and all the way up to Fe (K-edge: 7112eV; $K\alpha_1$: 6404 eV). The micro-X-ray beam was scanned over the sample (fixed beam position with sample moving) with a pixel size of ~5 × 5 and ~10 × 10 microns for VESPERS and SXRMB, respectively, while the intensity of the relevant fluorescent X-ray was tracked by silicon drift detectors. Elemental maps were produced with the normalized fluorescence yield of the element of interest and analyzed with standard imaging software.

XANES tracks the modulation of the X-ray absorption coefficient of an element above an absorption edge. The modulation, often appearing as oscillations arising from the interference of the forward- and backscattered photoelectron by surrounding atoms, contains all the information about the local structure of the absorbing atom (the absorption coefficient of a free atom is monotonic), such as the identity of the neighboring atom, its interatomic distance from the absorbing atom and its local dynamics (root mean displacement from the equilibrium distance). The simplest but most powerful use of XANES is its fingerprinting features since different local environments will exhibit different yet specific features in XANES. Thus, microbeam XANES provides additional information beyond XRF in that it provides not only elemental information but also chemical information. The power of these capabilities in the X-ray microanalysis of the Dzi bead specimen will become apparent below.

## 3. Results and Discussion

### 3.1. Optical and SEM Observations

The optical and SEM images are shown in Figure 2 together with the EDX analysis. A close inspection of the patterns revealed that in addition to the familiar eye and dark and light stripes, there were etched rings, as is apparent from Figure 1. Observation under the optical microscope did not

reveal any clear boundaries between the white ring and the dark background. Figure 2a displays the photographs of the one-eye and two-eye side of the bead together with the optical microscopy, and Figure 2b displays the SEM images showing a representative etched ring. These rings were all over the surface. In addition, from the slice in Figure S2, one could see the white region extended considerably into the bead; this confirms the report of Ebbinghouse and Winsten on the crafting of the white patterns that the white region extends significantly below the surface [2].

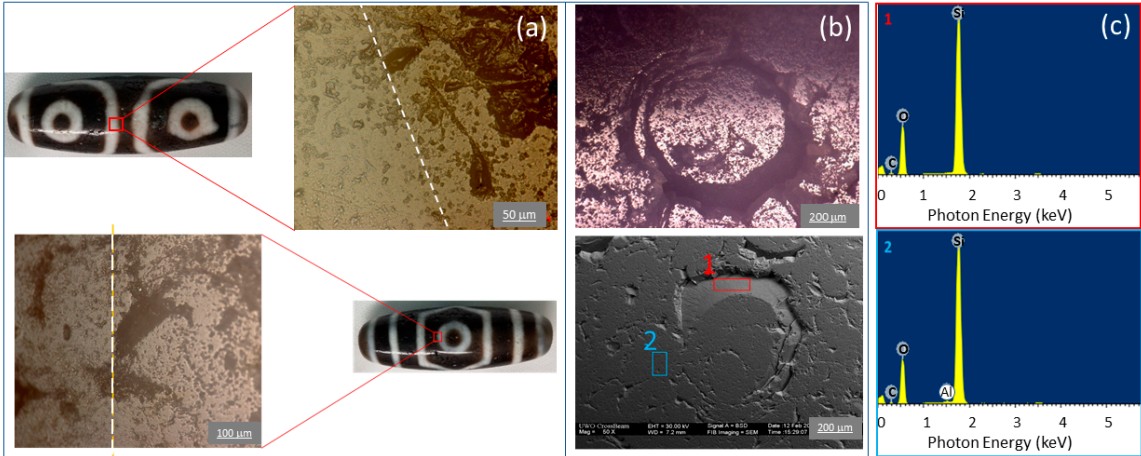

**Figure 2.** (**a**) The approximate boundary between the light and dark areas of the bead's pattern (depicted with a white dashed line), as observed using an optical microscope. (**b**) Optical (top) and scanning electron microscopy (SEM) (bottom) image of the etched rings that are all over the surface of the bead in both the light and dark regions and across the boundary. (**c**) EDX from two specified areas denoted 1 and 2 in the SEM (Electron High Tension (EHT): 30 kV). Dominant components are Si and O, as expected for agate.

One can also see from Figure 2b that the etched rings were about 1 mm in diameter and clearly exhibited cracks and crevices, as can be seen in both the optical and SEM images. EDX recorded in the selected area marked with blue and red rectangles identified Si and O as the dominant elements, with a tiny amount of Al in the flat region in Figure 2c. This finding immediately confirms the elemental composition of the Dzi bead was consistent with that of agate (quartz).

*3.2. XRD*

Figure 3 shows a comparison of the lab and synchrotron XRD recorded with (a) a powder sample of the Dzi bead and (b) the edge of a slice of the Dzi bead, respectively. The XRD using VESPERS was collected with a microbeam and an area-sensitive detector, yielding diffraction rings that were then converted to the 2θ display (Figure 1b and Figure S3) with the same scale as Co Kα for ease of comparison. From Figure 3a, one clearly sees that the Dzi bead powder exhibited a pattern characteristic of α-quartz (vertical lines), supporting the EDX results that the materials making up the Dzi bead were $SiO_2$ quartz. There was, however, a small but noticeable peak marked by the arrow. This conspicuous peak was not seen in the synchrotron data (Figure 3b), where a comparison was also made with the XRD of graphite and CuO. The latter had no match, e.g., CuO had strong peaks at 2θ between 30° and 40° (not shown), strongly indicating that this peak likely came from graphite. It should be noted that both graphite and CuO appear black.

From the above analysis, the Dzi bead was made of $SiO_2$ (quartz) and appeared to be genuine. A graphite signature was detected in the powder sample but not from microdiffraction at the edge of a slice; presumably, the latter probed a much smaller area than the powder sample examined with a large beam and a relatively macroscopic sample, although possible carbon contamination during the grinding process cannot be ruled out.

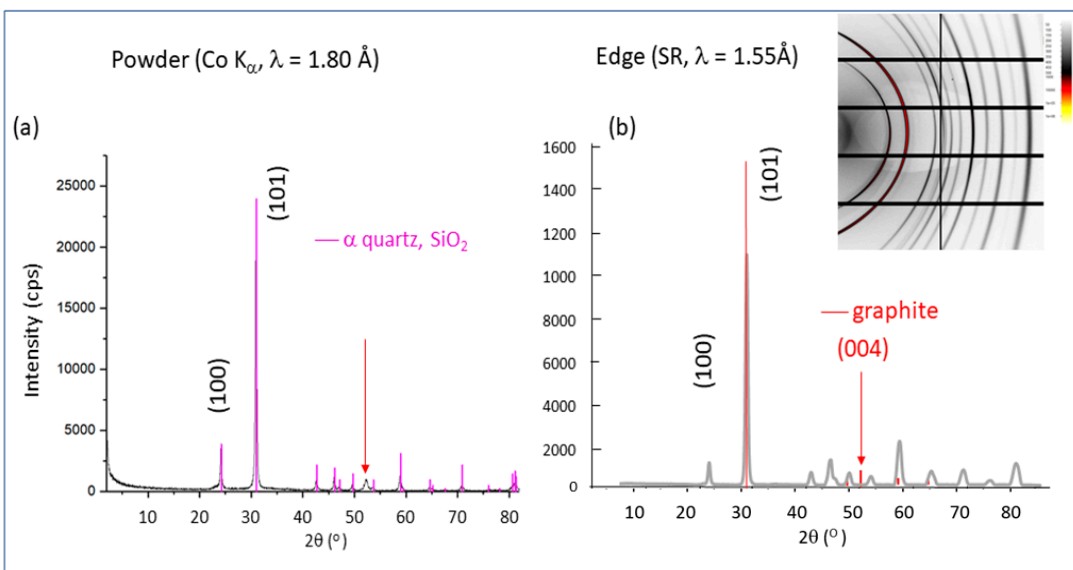

**Figure 3.** (**a**) Representative X-ray diffraction (XRD) of Dzi powder ground from a broken piece recorded using lab X-rays from a Co anode (λ = 1.8 Å). The corresponding XRD pattern of quartz (PDF 46-1045) is also shown in purple vertical lines. A conspicuous peak not belonging to quartz is marked with an arrow, which is consistent with graphite (004) diffraction. (**b**) XRD from the edge of a slice of Dzi bead obtained with 8 keV X-rays using the VESPERS beamline converted to the same 2θ scale as Co Kα for ease of comparison. The insert shows the diffraction rings from the Pilatus detector, which has been converted to the intensity-2θ pattern as shown. The XRD pattern of graphite (PDF 41-1487) is also shown as vertical red lines. The conspicuous peak seen in the powder sample is absent.

### 3.3. XRF, Imaging and Micro-XANES Analysis

Figure 4 shows the XRF map of elements in a region of interest covering both dark and light areas. XRF tracks the specific X-ray emissions from elements, e.g., Cu Kα X-ray emissions arise from the radiative de-excitation upon the removal of a 1s electron from a Cu atom (2p electron filling a 1s core hole, emitting an X-ray photon). While XRF can be excited by any high-energy radiation—most commonly electrons, as in SEM, EDX and X-rays—a synchrotron X-ray has the advantage over EDX using SEM in that, in addition to being many orders of magnitude brighter than laboratory X-rays, one can tune the excitation energy to just above an absorption edge of interest to enhance absorption and, hence, the intensity of the fluorescence or, under favorable conditions, to avoid an unwanted fluorescence signal by tuning the X-ray to below the edge of an unwanted element.

Let us first discuss the XRF results from VESPERS, a hard X-ray beamline where we used 12 keV X-rays, which will be able to excite Ca and all first-row transition elements, yielding characteristic $K_\alpha$ and $K_\beta$ X-rays; the $K_\alpha$ was used to track elements of interest.

From Figure 4a,b, we can see the etched rings covering the surface regardless of the dark and light regions. The scan covered an area of ~1.1 × 1.1 mm, as marked by the red square. It began in a dark region and ended in a light region, covering ~50–50 of both the light and dark region. From Figure 4c, it is apparent that the most interesting feature was the hot spots seen in the element maps of Ca, Fe and Cu, especially Cu. The location of these rings had no correlation with the dark and light regions of the surface. Let us now concentrate on the Cu map. The spots were ~20 μm in diameter and aligned regularly along the circumference of the ring, with a spacing of ~100 μm. The maps of Ca, Fe and Cu could be directly overlaid on each other. To determine the chemical composition of the Cu, we conducted Cu K-edge XANES measurements with a microbeam at the hot spots. Figure 5a,b shows a magnified image of this region, which reveals that the etched ring spanned both the light and dark regions. Figure 5c displays the micro-XANES of a hot spot (5 × 5 μm) recorded in the Cu $K_\alpha$

fluorescence yield compared with the XANES of the reference compounds, Cu metal, $Cu_2O$ and CuO, representing the 0, +1 and +2 oxidation states of Cu. It should be noted that despite low count rates in micro-XANES, the signal we obtained (eight scans) required about an hour, and it already clearly confirms that the hot spot Cu was from CuO. The discovery of CuO led us to compare the XRD data with those of CuO XRD patterns, and there was no sign of any CuO peaks seen in the bead's data, as noted in the XRD section above. This indicates that the amount of CuO was either amorphous or very small and not detectable in the XRD.

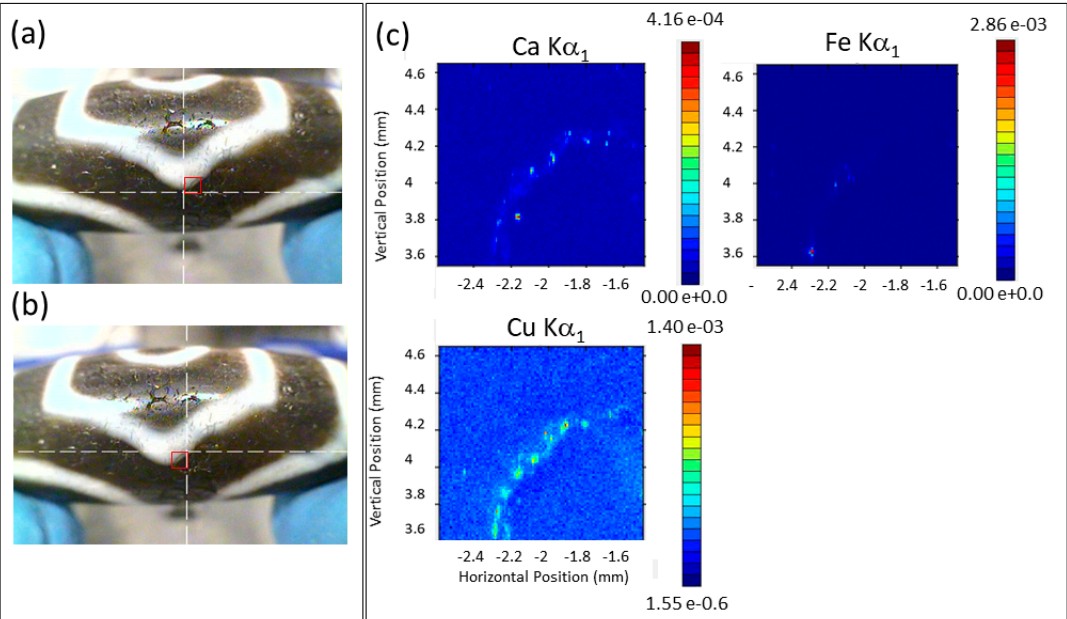

**Figure 4.** (**a**,**b**) show the starting and ending point of the scan (crosshairs), respectively, of the region marked with a square. (**c**) X-ray fluorescence (XRF) maps of several elements, Ca, Fe and Cu, are shown (XRF data are stored in a Multichannel Analyzer (MCA); counts within an energy window characteristic of the element are tracked and color-coded). These maps show hot spots on the circumference of the etched ring, most apparent in the case of Cu. The size of the hot spot is ~20 μm, with ~100 μm spacing along the circumference.

While the experiment using VESPERS provided fruitful results, its energy was not suitable to track $SiO_2$, the major component of agate based on which Dzi beads are made and crafted. We next conducted measurements with the SXRMB beamline using its tender X-ray microprobe endstation. SXRMB is designed for tender X-ray (1.7–5 keV) investigations, with accessible photon energy extending to 10 keV [16,17].

The microprobe provides a beam size of ~11 × 11 μm across this energy range with KB mirror focusing [17]. To investigate the Si K-edge, we used InSb(111) crystals in the double-crystal monochromator. XRF, imaging and micro-XANES were all conducted in a vacuum chamber; a Si drift detector was used to track the fluorescent X-rays. Figure 6a shows the optical image of the area of interest. Figure 6b displays the XRF maps excited at a photon energy of 7130 eV, just above the Fe K-edge, and selected regions of interest (ROI), marked with color-coded boxes. Figure 6c–e is the XRF maps of Si, K, Ca and Fe of selected ROI, respectively. While the etched rings were apparent, instead of looking more like holes in the Cu map, the Si in Figure 6a shows the optical image from the video camera; it again shows that the etched rings were all over the surface. The boxes define the area where XRF maps of relevant elements were collected, as shown in Figure 6b, from Si, K, Ca and Fe $K_{\alpha 1}$ emissions. It can be clearly seen that the rings contained extruding Si materials, while other elements appeared to be less defined or depleted in the case of K, which tracked Si well in the ring circumference. It was desirable to see if the Si map was correlated with the Cu image obtained using VESPERS (Figure 5). While Cu mapping using SXRMB was difficult due to low flux at the Cu K-edge, we were

able to track the Cu distribution while tracking Si and other element maps at 7130 eV by observing the Cu K$_\alpha$ produced by second-order radiation (14260 eV). The results are shown in Figure S4 for ROI displayed in Figure 6d; one can see that despite the weak signal, there was a good correlation between Cu and Si.

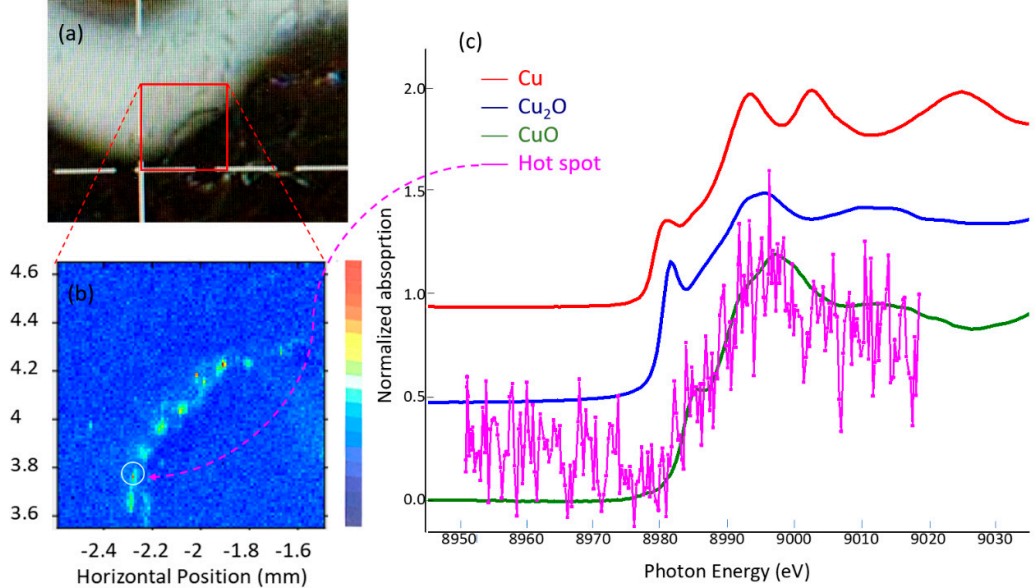

**Figure 5.** (**a**) Optical image of the magnified region of the etched ring inside the region of the XRF map. (**b**) XRF map from Cu showing the hot spots and the circled hot spot where micro-X-ray absorption near edge structure (XANES) has been obtained. (**c**) Cu K-edge micro-XANES from a hot spot circled in (**b**), obtained with a microbeam (5 × 5 μm) together with the references, Cu metal, Cu$_2$O and CuO.

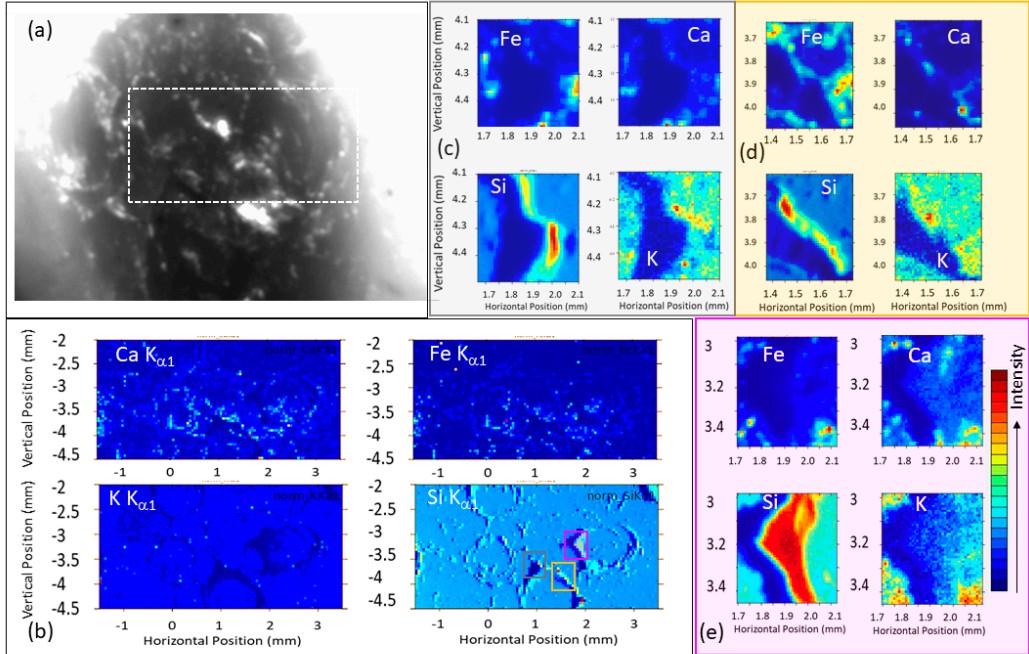

**Figure 6.** (**a**) Optical image of the Dzi bead surface; the rectangle encloses the region of the XRF maps shown in (**b**). (**b**) XRF map from Ca, K, Si and Fe as noted; the color-coded boxes in the Si map define the regions of interest (ROI) displayed in (**c**–**e**) with the same color code, where all four elements are tracked; the color code is also shown.

Figure 7 displays the Si K-edge micro-XANES of four spots from hot to cold. We can see that the XANES for all the spots was nearly identical except for spot 2, which was at the edge of the crack and

showed a slight broadening of the resonances, indicating some disorder. They were characteristic of the XANES of quartz (inset), with the white line (spike at the edge jump) at ~1848 eV, arising from the Si 1s to $t_2$ transition in the tetrahedral surrounding of Si in quartz. We have also conducted micro-XANES in different ROIs depicted in Figure 6c,d and found similar results (not shown) with no irregularities.

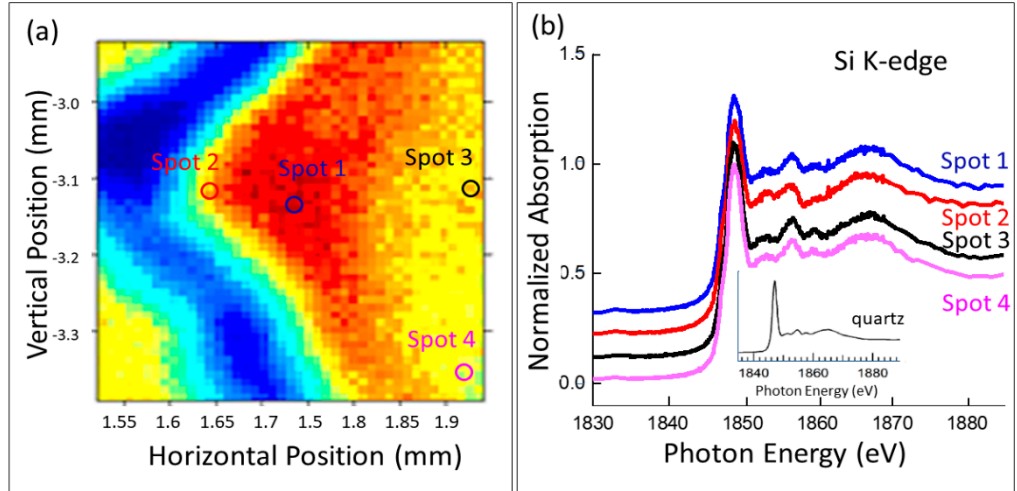

**Figure 7.** (**a**) Si map of a ROI of the Dzi bead sample, as shown in Figure 6e, where four spots (single pixel of $11 \times 11$ μm) were selected for micro-XANES analysis as marked. Spot 1 is the hot spot, spot 4 is the cold spot, spot 2 is at the edge of a crack and spot 3 is with moderate intensity. (**b**) Si K-edge XANES of selected spots. The spectral features are unmistakably characteristic of quartz $SiO_2$ (inset).

## 4. Discussion

We have presented and analyzed all data from XRD, XRF imaging and micro-XANES of selected regions of the Dzi bead. We now attempt to understand what these results mean in connection with the questions we set out to explore: (1) what is this three-eye Dzi bead made of, (2) is it genuine or fake and (3) how was the crafting done?

We begin with the first question. Based on the XRD, XRF mapping and XANES, there is absolutely no doubt that this sample was made of $SiO_2$ (quartz) consistent with an agate origin. For the second question, since it is generally agreed by Dzi bead collectors and traders that Dzi beads are made of agate, our observations clearly show that it was made of $SiO_2$ (quartz), which eliminates all the fake possibilities illustrated in Figure 2c (plastic with metal core, polymer clay and painted aluminum).

The third question is perhaps the most interesting yet challenging and is most relevant to our analysis. Based on the descriptions of Bolin [1] and Ebbinghouse and Winsten [2], there are many ways to inspect Dzi beads and how they are crafted. We will focus on the surface-etched rings; the dark and light pattern; and how the XRF map, especially the Cu pattern we found in this analysis, may reveal how this three-eye Dzi bead was crafted.

From the optical image, we see that it showed well-defined light (white) and dark patterns, and the surfaces were covered with etched rings; these are the general characteristics of Dzi beads. Upon slicing the bead in half, we could see that the piece was translucent, and the white pattern did extend considerably into the bulk, as described by Ebbinghouse and Winsten [2]. None of the SEM, EDX, XRD and XRF mapping techniques showed any clear boundary between the light and the dark region, however. XRD of the powder showed the presence of graphite, while microdiffraction of the edge of the slice did not. It is unclear whether this is due to the sensitivity of the microbeam looking at a very small area or whether the powder was contaminated by carbon during the grinding process. This awaits future exploration. The presence of graphite may be related to light absorption in the dark region, however.

The most interesting finding was the hot spots of Cu on the circumference of the etched ring. In both Bolin's book [1] and the work of Ebbinghouse and Winsten [2], drilling with small bits and

the use of alkaline material and copper nitrate, $Cu(NO_3)_2$, were mentioned as agents put on the bead surface, followed by firing for whitening and darkening functions, respectively. The regular separation (~1 mm) of the Cu spots of ~20 μm in size could be speculated to be the tiny drill holes to facilitate diffusion of chemicals used in the pattern formation. The presence of CuO in all the Cu hot spots was likely the result of the following reaction upon firing:

$$Cu(NO_3)_2 \text{ (s)} \rightarrow CuO \text{ (s)} + N_2 \text{ (g)} + 3O_2 \text{ (g)}$$

In addition to Cu, Ca and Fe were also found with both 12.0 keV (VESPERS) and 7.13 keV (SXRMB) excitation photon energy in the XRF map overlaying that of the Cu. K was also found in the tender X-ray map (not looked for in VESPERS); the origin of Ca is less clear, and Fe may have come from the tools used in crafting. It should be noted, however, that Fe is also commonly found in agate as a trace element, while Ca and K are not [18]. Compared to the Si map, which showed extruding pilings of $SiO_2$ on the circumference of the rings, the Cu spots appeared to be more like tiny pits. The Si and Cu map were correlated (Figure S4. It is also interesting to note that while we found CuO from the micro-XRF, it could not be detected in the XRD.

Finally, our preliminary finding was consistent with the descriptions noted in the literature [1,2]. Presumably, a piece of translucent agate was polished into shape and treated with alkaline solution. $SiO_2$ is slightly acidic, and a potash solution could do the etching (dissolving $SiO_2$), and the light region could come from the increased porosity. The use of pyrolytic $Cu(NO_3)_2$ and firing would produce CuO, which appears black. The origin of regular hot spots of CuO on the ring was possibly from drilling. The presence of graphite remains a mystery; repeated XRD measurements of various regions of the bead (not powder) revealed no crystalline carbon. The origin of carbon requires further investigation, although if plant sugar was used for darkening under heat and reed drill bits were used, it could be a source of carbon.

## 5. Conclusions

We have reported the preliminary results of an exploration of the mysterious Dzi beads by studying a three-eye bead using laboratory-based optical microscopy and SEM and the advanced synchrotron techniques including XRD, XRF imaging and micro-XANES. The specimen, including the surface, the edges and the interior after slicing, was inspected with an optical microscope and SEM. The crystal structure from a powder specimen, the surface and the edges of a sliced piece were subjected to XRD. They all showed the structure of quartz ($SiO_2$). While there appeared to be a trace of graphite in the former, it was not detected in the latter. XRF imaging also shows that the bead was made of $SiO_2$; in addition to $SiO_2$, there were regular Cu hot spots, as well as Fe, Ca and K at lower concentrations on the circumference of the etched rings. We can conclude, from the material perspective, that the three-eye bead specimen was genuine, and the patterns (dark and light as well as the etched rings), which were apparent to the naked eye, were crafted. The discovery of regular small pits containing copper oxide (CuO) on the circumference, which correlated with the $SiO_2$ map, and the presence of potassium were consistent with the crafting methods described in the literature [1,2].

Finally, while we have not fully answered the question on the origin of these beads nor expect to do so with a single specimen, in a broader context that will require further studies of representative samples from various regions and times, we are optimistic that the tools and the methodologies introduced here will play a significant role in future studies.

**Supplementary Materials:** The following are available online at http://www.mdpi.com/2571-9408/3/3/56/s1, **Figure S1**. The three-eye Dzi beat purchased from a jewelry piece came with an authenticity certificate and a bag. **Figure S2**. (a) The slicing plan for the Dzi bead as seen lengthwise. (b) A depiction of the Dzi bead slicing plan as seen from one end of the bead, looking down the hole that runs lengthwise through the bead. (c) Dzi bead post-slicing. (d) Dzi bead after ultrasound bath. **Figure S3**. Glazing XRD (left panel) of the surface light and dark regions (red dot on the right panel) using 12 keV (1.03 Å) X-rays. Despite the intensity variation, the patterns are identical. **Figure S4**. Cu Kα map recorded based on second order radiation at the Fe K-edge excitation compared with corresponding maps for Si, Fe and Ca.

**Author Contributions:** A.R. conducted most of the lab measurements at Western; R.F. and T.-K.S. conducted measurements at the VESPERS and Q.X., Y.H. and T.-K.S. conducted measurements at the SXRMB; T.-K.S. initiated the project and provided funds for the research. All authors participated in data analysis and discussion. All authors have read and agreed to the published version of the manuscript.

**Funding:** This research was supported by the Canada Research Chair (CRC) program and a Natural Science and Engineering Research Council (NSERC) Discovery Grant. Research was in part conducted at the Canadian Light Source which is supported by Canada Foundation for Innovation (CFI), National Research Council (NRC), Canadian Institute of Health Research (CIHR), NSERC, and the University of Saskatchewan.

**Conflicts of Interest:** The authors declare no conflict of interest.

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
