# Peer review of "Exploring the Dzi Bead with Synchrotron Light: XRD, XRF Imaging and μ-XANES Analysis"

_heritage, doi:10.3390/heritage3030056_

Round 1
Reviewer 1 Report
Interesting introduction and exciting topic. Well developed research project. Clear and understandable presentation. Methods and results adequately exposed. Understandable language. Exhaustive bibliography
Author Response
We thank the reviewer for the positive comments
Reviewer 2 Report
It is an interesting work, in which a Dzi bead -stones of relevant cultural importance for people in Tibet and sorrounding areas - has been characterized with x-rays.
although only one Dzi bead has been analyzed and the study is not conclusive to explain the origin of the patterns visible on the surface of these stones, the paper deserves to be published. it is an useful preliminary work to realize further studies and more accurate characterization of Dzi beads to clarify definitively the origin of the patterns visible on them.
However, some hints have to be addressed.
everywhere in the paper words such as mysterious and mystery have to be avoided.
It is more scientific the use of words such us "unkwown","not yet explained", and so on.
minor checks...
Lines 23-24 -the last sentenceof the abstract "the prospect of synch..." is superfluous, and can be neglected.
Line 95:the meaning of R&D has not been specified.
lines 383-385. the last sentence is a typo and should be erased.
The vertical legend of Figure S4 should be corrected.
Reviewer 3 Report
In general, I think this is a useful article but needs some revisions and re-focus in order to be of interest to a wide audience. To my understanding the analytical approach is sound; however, the paper fails to highlight the cultural significance of the findings. For example, the stories discussed in the introduction are interesting but we are missing the archaeological component - do we have this category of objects in archaeological sites? where are they found? how are they deposited? These are important questions that need to be addressed if the paper is to be of use in heritage studies.
We are told in the abstract that the origins of the bead will be addressed - the paper does not really do this, except to tell us what the raw material of the bead is. However, I would like to read more about the potential origins of these beads. Are there agate sources in Tibet? Has the bead originated elsewhere completely? The analyses do not really add anything to the discussion of the origin (geographical or age) of the bead - some comparisons with geological samples of agate would be interesting but time consuming and probably require another dedicated paper.
It would, also, be interesting to read more about the history of these beads in greater detail - can we make any cultural inferences from them? Can use these beads as a proxy of long distance trade, communication networks in different points in time? These are interesting questions and issues that I feel need to be addressed in the text to highlight the significance of studying this type of objects and the application of such innovative techniques.
I would argue that the opening statement in the abstract regarding the origin of the analysed bead is not actually realised via the research. I would suggest the focus of the paper changes to reflect what we actually learn about the bead from these analyses, this being how the patterns on them are made (i.e. manufacture processes). This change in the paper's focus will help tie everything better together.
Another important concern regarding this paper is how much can we actually learn from a single object?How does one bead help the overall understanding of the beads heritage value? In the same context, why it is important to know if a single shop-bought bead is genuine or not? What is important about this bead? Some elaboration regarding the specific bead is required to help the reader understand and evaluate the significance of this object.
Finally, regarding the final introductory statement (line 24) of the paper: indeed, synchrotron techniques can contribute crucial new data to heritage research. Nevertheless, it needs to be emphasised that the use of such destructive techniques can only have limited application to cultural remains. This is a major limiting factor in the application of synchrotron methods on cultural materials.
In short, I think it is important to test novel techniques to the study of material remains of the past but the results must be interpreted through a cultural - humanistic lenses if we are to elucidate aspects of our past/heritage.
Reviewer 4 Report
The work by Dr Reinhardt et al is original and interesting, as it regards special Tibetan beads rarely studied by the scientific community. The manuscript is overall well written and encourages the reader to elaborate on the topic. The use of top quality analytical methods guaratees the reliability of the study itself and it surely deserves pubblication.
I have just minor comments/revisions to suggest:
- The composition of agate is not my area of expertise, but I wonder whether Ca, Fe and K could be trace elements in this rock. Was this hypothesis taken into account?
- I would suggest to add references not only by the author's reasearch group, but also something more general on other synchtrons. There are indeed several facilities worlwide working (also) on cultural heritage applications, the work of which may contextualize this work in the Heritage Science field
- Would it be possible to adjust the contrast in fig 4c in the Fe, to ease the reading of the elemental distribution?
There are also few typos:
- References in text are sometimes in square brackets and sometimes superscript
- lines 226 and 227 power instead of powder
- lines 236-237: the phrase "while XRF..." seems incomplete
- lines 364-366 and 383-385: left of journal suggestions
- ref 15 is incomplete
Round 2
Reviewer 3 Report
The revised manuscript is definitely improved. Although I would have liked more archaeological/cultural context I realise this is not the focus of the paper or the authors' expertise. The revisions, especially in the conclusions section, clarify earlier issues and put the presented work in a better context to be understood and evaluated by humanities researchers too.